# Effects of Sorghum Grain and Wort Composition on Dry Grind Fermentation Performance: A Model for Baijiu Production

Siong H. Tan [1], Christopher L. Blanchard [1], Thomas H. Roberts [2,3,*], Daniel L. E. Waters [1,†] and A. John Mawson [1,‡]

1   Gulbali Institute, Charles Sturt University, Wagga Wagga, NSW 2650, Australia
2   Plant Breeding Institute, School of Life and Environmental Sciences, University of Sydney, Camperdown, NSW 2006, Australia
3   Sydney Institute of Agriculture, University of Sydney, Camperdown, NSW 2006, Australia
*   Correspondence: thomas.roberts@sydney.edu.au
†   Current address: Faculty of Science and Engineering, Southern Cross University, Lismore, NSW 2480, Australia.
‡   Current address: The New Zealand Institute for Plant & Food Research Limited (Plant & Food Research), Ruakura Research Centre, Hamilton 3214, New Zealand.

**Abstract:** Sorghum grain is the principal raw material for Baijiu production, but the effects of grain and wort composition on fermentation performance are unclear. Ethanol production at laboratory scale using grains of 11 commercial sorghum cultivars from a field trial was investigated using dry grind fermentation. Initial wort glucose content was 141–150 g/L and fermentability (glucose-to-ethanol conversion rate) was 87–90%. Ethanol production rate among sorghum genotypes ranged from 1.18 to 2.04 mL of ethanol per litre wort per hour of fermentation. The cultivars were categorised into four groups according to a fermentation endpoint of 60–69 h, 70–79 h, 80–89 h and >90 h. All but one of the sorghums produced a final ethanol content of 9.47–9.76% *v/v*. Cultivars with high-starch and low-protein grains were the most suitable for fermentation due to the high final ethanol content and fermentability achieved. Initial wort glucose content and yeast assimilable nitrogen content were not correlated with grain starch content, protein content, ethanol content, fermentability, ethanol production rate or glucose consumption rate. Knowledge of the effects of sorghum grain quality on fermentation performance can pave the way for further research to optimise solid-state fermentation for Baijiu production.

**Keywords:** baijiu; ethanol; fermentation; glucose; *Sorghum bicolor*; starch; wort

## 1. Introduction

Baijiu is a clear and colourless Chinese spirit distilled mainly from fermented sorghum, although other grains such as rice, barley, millet and maize may be included in the solid-state fermentation [1]. Baijiu contains a high ethanol content ranging from 38 to 65% [2] and is mostly consumed neat. Based on the record of a distiller from the tomb of Haihunhou, the fermentation of sorghum to produce Baijiu has been practised in China for two millennia, dating from the Western Han dynasty (from 202 B.C. to 8 A.D.) [1]. The high level of flavour complexity of Baijiu is due to a myriad of compounds produced by the microbial community responsible for the fermentation process [3].

The global alcoholic drinks market size was valued at USD 1448 billion in 2021 and is expected to expand at a compound annual growth rate of 10.3% from 2022 to 2028, according to Grand View Research, a market research and consulting company [4]. In this market, the world's top four most valuable spirits brands were found to be Baijiu producers, namely Moutai, Wuliangye, Yanghe and Luzhao Laojiao [5]. The economic importance of Baijiu supports research to better understand the properties of sorghum grain that influence fermentation performance.

Carbohydrates present in grain sorghum are both structural, including cellulose and pectin, and non-structural, including sugars and starch [6,7]. For ethanol production, starch and cellulose must be hydrolysed into fermentable sugars. Different sorghum cultivars have different starch contents, leading to a range of ethanol content in the fermented product. Moreover, differences in yeast assimilable nitrogen (YAN) content of the wort result in different fermentation rates.

The aim of this study was to investigate the fermentation performance of grains from a selection of 11 sorghum cultivars (grown in a field trial in Australia) for ethanol production. A well-defined standard dry-grind method was employed as the complexity and diversity of the traditional Chinese Baijiu production system make it difficult to establish a reproducible fermentation system outside China that mimics the traditional Chinese fermentation processes. The fermentation performance of each cultivar, including changes in ethanol, glucose and YAN content, as well as pH and specific gravity, was determined and correlated to major grain and wort compositions. The significance of this study is that it establishes relationships between quality parameters of non-waxy, low-tannin sorghum grain and dry grind fermentation performance. The results can be used to inform research to optimise grain quality for solid-state fermentation in Baijiu production.

## 2. Materials and Methods

### 2.1. Grain Samples

Sorghum (*Sorghum bicolor* L. Moench) grains of 11 Australian cultivars were obtained from field trials conducted at Narrabri, NSW (30.3324° S, 149.7812° E) in 2016/17, by the University of Sydney Plant Breeding Institute. Seeds of four cultivars from Nuseed (Rippa, Liberty, Cracka and Tiger), three cultivars from Pioneer (84G22, 85G33 and 85G44) and four cultivars from Pacific Seeds (MR-Apollo, MR-Bazley, MR-Buster and MR-Taurus) were planted in October 2016 and harvested in March 2017. These cultivars are all non-waxy, have very low tannin content (<0.25% catechin equivalent) and have different field performance such as harvest yield, kernel weight and grain size [8,9].

### 2.2. Analysis of Sorghum Grains

The moisture content of the whole-grain sorghum was measured using an Infratec 1241 Grain Analyzer (Foss, Hilleroed, Denmark) [8]. Total starch content was analysed using a total starch kit (K-TSTA-100A) purchased from Megazyme (Wicklow, Ireland) (https://megazyme.com (accessed on 1 February 2018)) according to the manufacturer's instructions. The protein content of the samples was determined by Kjeldahl Method 954.01 with a conversion factor of 6.25.

### 2.3. Dry Grind Fermentation

Fermentation of sorghum under dry grind conditions (liquid fermentation) was carried out according to the method of Wu et al. (2007) [10] with modifications. Whole sorghum grains were ground to flour (500 μm) using an Ultra Centrifugal Mill ZM 200 (Retsch, Haan, Germany). Water (450 mL) was added to the flour (135 g) to form a slurry. A 45 μL aliquot of an α-amylase blend (Spezyme® AA, Croydon South, Australia) containing 6500 thermostable amylase units (TAU)/g from The Enzyme Solutions was added and the slurry heated in a shaking water bath (97 °C, 120 rpm). The temperature was held at 97 °C for 45 min, then lowered to 80 °C. A second 45 μL aliquot of amylase blend was added. The mixture was held at 80 °C for 30 min, then cooled to 60 °C and 450 μL amyloglucosidase (260 units/mL) from Sigma-Aldrich (Castle Hill, NSW, Australia) was added. The temperature was held at 60 °C for 30 min, then lowered to room temperature. HCl (1 M) was used to adjust the pH of the slurry to 4.1–4.3.

Yeast (*Saccharomyces cerevisiae* ex. *bayanus*; Lalvin EC-1118; 4.5 μg) (Lallemand, Montreal, Canada) was rehydrated in 15 mL deionised water at 40 °C for 20 min and then added to the slurry. The total mass of the mixture was adjusted to 665 g using deionised water and then mixed well for 2 min. A day 0 sample (50 g) was collected and kept in a

refrigerator (ca. 4 °C) (Electrolux Home Products, Mascot, Australia) until analysis. The inoculated slurry was fermented in an incubator (30 °C) with sample collection every 12 h for a total of 96 h. The samples collected were centrifuged (3000 rpm for 10 min) to obtain clear supernatants, which were then filtered by Whatman qualitative filter paper Grade 1 before analysis.

### 2.4. Analysis of Fermented Samples

Ethanol content was determined by Near Infrared Spectroscopy using an Anton–Paar Alcolyzer (Anton Paar, Graz, Austria), as described by the European Brewery Convention (EBC) method 9.2.6. Ethanol production rate was calculated as the slope of the ethanol curve between 24 and 48 h. Fermentation endpoint was determined as the time point at which the wort reached maximum ethanol content.

An Anton–Paar Density Meter DMA 4500 (Anton Parr, Graz, Austria) connected to the Alcolyzer was used to determine sample specific gravity. Glucose content was analysed via a Thermo Scientific Arena 20XT Analyzer using a colorimetric test with glucose oxidase/peroxidase reagent kit (984304, Thermo Scientific, Scoresby, Australia). The glucose depletion rate was calculated as the slope of the glucose curve between 24 and 48 h. A CyberScan pH 510 (Eutech Instruments, Singapore) was used to determine sample pH values. The rate of pH decrease was calculated as the slope of the pH curve between 24 and 48 h.

Ammonia content was analysed using a colorimetric test with a glutamate dehydrogenase reagent kit 984320 (Thermo Scientific, Scoresby, Australia), while $\alpha$-amino acid content was determined using a colorimetric test with an *o*-phthaldialdehyde and *N*-acetylcysteine reagent kit 984342 (Thermo Scientific, Scoresby, Australia). Both tests were carried out using an Arena 20XT Analyzer (Thermo Scientific, Scoresby, Australia). YAN content was determined using the following equation:

$$\text{YAN} = 0.8225 \times \text{ammonia content} + \alpha\text{-amino acid content} \tag{1}$$

Fermentability of each standardised dry grind fermentation sample was calculated based on the standard method of the Institute of Brewing (IOB) as described by Bathgate (1981) [11]:

$$\text{Fermentability (\%)} = \frac{\text{OG} - \text{FG}}{\text{OG} - 1.000} \times 81.9 \tag{2}$$

where OG = specific gravity before fermentation (0 h) and FG = specific gravity after fermentation (96 h).

### 2.5. Statistical Analysis

All extractions and analyses were carried out three times and the means with standard deviation reported. Data collected were subjected to analysis of variance (ANOVA) using SPSS™ statistical software version 25 (SPSS Inc., Chicago, IL, USA). Fisher's least significant difference (LSD) test was used to identify the means that were significantly different ($p < 0.05$). Correlations between fermentation performance and grain and wort major compositions were obtained by calculating Pearson's correlation coefficient (r).

## 3. Results and Discussion

### 3.1. Major Grain Composition

Long-term storage of sorghum grain requires a moisture content less than 13.5% [12]. The moisture content of the sorghum grains here was consistently at 11.3–11.9% (Table 1). Thus, all sorghum samples had the normal range of moisture suitable for grain storage (~12%) at the time of harvest.

**Table 1.** Major composition of grains of 11 commercial sorghum cultivars.

| Cultivar | Grain Moisture Content (%) | Starch (%) | Protein (%) |
|---|---|---|---|
| MR-Apollo | 11.89 ± 0.42[a] | 58.26 ± 0.76[d] | 12.90 ± 0.09[a] |
| Rippa | 11.80 ± 0.25[ab] | 59.48 ± 1.04[bd] | 12.40 ± 0.29[ab] |
| MR-Bazley | 11.73 ± 0.27[ab] | 60.46 ± 2.28[bcd] | 11.68 ± 0.31[cd] |
| MR-Buster | 11.71 ± 0.19[ab] | 60.78 ± 0.44[abcd] | 11.74 ± 0.31[cd] |
| 84G22 | 11.85 ± 0.34[ab] | 61.03 ± 1.46[abcd] | 11.52 ± 0.35[c] |
| 85G33 | 11.92 ± 0.26[a] | 62.31 ± 1.99[abc] | 11.53 ± 0.23[c] |
| Liberty | 11.26 ± 0.16[b] | 63.63 ± 2.18[ac] | 11.43 ± 0.26[ce] |
| MR-Taurus | 11.90 ± 0.41[a] | 59.04 ± 1.90[bd] | 12.25 ± 0.17[bd] |
| Cracka | 11.90 ± 0.18[a] | 60.39 ± 0.98[bcd] | 11.71 ± 0.28[cd] |
| 85G44 | 11.83 ± 0.28[ab] | 63.99 ± 0.15[a] | 10.88 ± 0.23[e] |
| Tiger | 11.43 ± 0.15[ab] | 59.59 ± 1.45[bd] | 12.62 ± 0.45[ab] |

Different letters within each column indicate significant difference in value ($p < 0.05$).

The starch content of the sorghum grains was in the range 58.3–64.0% (Table 1). This is consistent with a study which reported a starch content of 62.4–64.9% among Australian cultivars [9]. A larger range of values for starch content (59.0–72.5%) was reported in another Australian study on grains from 55 sorghum genotypes [13]. Wu et al. (2007) studied 70 sorghum genotypes and elite hybrids in the U.S. and reported starch content of 64.3–73.8% [10].

Protein content of the sorghum grains tested was in the range 10.9–12.9% (Table 1). This is consistent with the protein content of 11.0–12.1% reported by Ramírez et al. (2016) [14]. Among the 11 cultivars tested in the current study, the protein content of MR-Apollo, Rippa, MR-Taurus and Tiger was more than 12%, while 85G44 was the only cultivar with a protein content less than 11%. Grain protein content was strongly and negatively correlated to starch content (r = −0.896, $p < 0.001$), consistent with the findings of Sailaja (1992) [15].

### 3.2. Fermentation

#### 3.2.1. Production of Ethanol

The production of ethanol by yeast during fermentation followed the trend observed for a typical yeast growth curve [16]. There was a lag phase in ethanol formation during the initial 12 h of fermentation, followed by a log phase, then a short deceleration phase before a stagnant phase at the end of the 4-day fermentation (Figure 1a). A stagnant phase was not obvious for 85G44 and Tiger due to the slow fermentation process. All other cultivars reached the fermentation endpoint in less than 90 h (Table 2).

MR-Apollo had the lowest ethanol content (8.94% *v/v*) after 96 h of fermentation while the ethanol content of all the other cultivars was 9.26–9.76% *v/v* (Table 2). Mamma et al. (1995), who carried out liquid fermentation of sweet sorghum stalk, reported a maximum ethanol content of 9.2% *w/v* (11.7% *v/v*) after 48 h [17]. Based on the fermentation endpoint, the sorghum cultivars tested here were categorised into four groups according to a fermentation endpoint of 60–69 h (MR-Apollo and Rippa), 70–79 h (MR-Bazley, MR-Buster, 84G22, 85G33, Liberty and MR-Taurus), and 80–89 h (Cracka) and ≥90 h (85G44 and Tiger).

Our results show that it is possible to carry out sorghum fermentation at 10–12% solids and still obtain a commercially acceptable ethanol content of 8–9% *v/v* [18]. In their study on co-fermentation of sorghum grain meal and stem juice at 13–14% solids (expressed as sucrose content), Bvochora et al. (2000) reported very low final ethanol content (3–4.5% *v/v*) [19]. Another set of experiments carried out by the same group resulted in a final ethanol content of 11–16.8% *v/v* but at 34% solids. Distillation of fermented products with about 10% *w/v* ethanol is optimal for cost efficiency with little cost savings beyond that level [20].

**Table 2.** Fermentation endpoint, initial (0 h) and final (96 h) wort composition and rate of changes, pH and specific gravity of 11 commercial sorghum cultivars.

| Cultivar | Fermentation Endpoint (h) | Ethanol Content (% *v/v*) | | Ethanol Production Rate (ml/L h) | Glucose Content (g/L) | | Glucose Consumption Rate (g/L h) | Fermentability (%) |
|---|---|---|---|---|---|---|---|---|
| Fermentation Time (h): | | 0 | 96 | 24–48 | 0 | 96 | 24–48 | 96 |
| MR-Apollo | 62.4 ± 0.8[a] | 0.26 ± 0.06[a] | 8.94 ± 0.00[f] | 2.03 ± 0.06[a] | 141.32 ± 3.38[i] | 0.42 ± 0.13[ab] | 3.41 ± 0.06[a] | 87.33 ± 0.20[h] |
| Rippa | 66.8 ± 5.8[a] | 0.17 ± 0.01[bc] | 9.47 ± 0.01[d] | 1.93 ± 0.03[ac] | 148.35 ± 2.61[ac] | 0.43 ± 0.10[ab] | 3.08 ± 0.01[bc] | 89.05 ± 0.67[bcd] |
| MR-Bazley | 75.8 ± 1.4[b] | 0.18 ± 0.06[bd] | 9.60 ± 0.05[bd] | 1.79 ± 0.02[bd] | 144.65 ± 0.49[ci] | 0.33 ± 0.10[ab] | 2.91 ± 0.06[de] | 88.60 ± 0.42[b] |
| MR-Buster | 73.6 ± 0.7[b] | 0.16 ± 0.01[bd] | 9.59 ± 0.02[ed] | 1.98 ± 0.01[a] | 152.85 ± 0.96[ad] | 0.53 ± 0.05[ab] | 3.22 ± 0.11[b] | 88.42 ± 0.26[bd] |
| 84G22 | 78.9 ± 5.5[b] | 0.13 ± 0.08[b] | 9.76 ± 0.19[ab] | 1.79 ± 0.07[bf] | 144.38 ± 3.00[ci] | 0.36 ± 0.16[a] | 2.83 ± 0.06[dg] | 89.07 ± 0.15[be] |
| 85G33 | 74.5 ± 0.6[b] | 0.17 ± 0.01[bd] | 9.59 ± 0.09[dg] | 1.87 ± 0.03[cdf] | 144.06 ± 2.48[ei] | 0.42 ± 0.22[ab] | 2.96 ± 0.10[cegh] | 89.03 ± 0.21[bf] |
| Liberty | 74.9 ± 2.1[b] | 0.18 ± 0.01[bd] | 9.73 ± 0.07[beg] | 1.76 ± 0.10[b] | 146.22 ± 1.87[cef] | 0.74 ± 0.49[ab] | 2.83 ± 0.07[dh] | 89.48 ± 0.64[cdefg] |
| MR-Taurus | 79.0 ± 5.6[b] | 0.21 ± 0.01[acd] | 9.55 ± 0.03[d] | 1.63 ± 0.04[e] | 147.42 ± 3.11[ceh] | 1.02 ± 0.84[ab] | 2.66 ± 0.17[f] | 89.24 ± 0.11[bd] |
| Cracka | 86.2 ± 0.5[c] | 0.16 ± 0.04[bd] | 9.56 ± 0.11[d] | 1.54 ± 0.04[e] | 146.01 ± 1.83[ceg] | 0.50 ± 0.23[ab] | 2.52 ± 0.11[f] | 89.02 ± 0.28[abg] |
| 85G44 | ≥96[d] | 0.22 ± 0.04[acd] | 9.75 ± 0.03[be] | 1.25 ± 0.03[g] | 150.50 ± 1.64[abfgh] | 0.74 ± 0.00[b] | 1.98 ± 0.01[i] | 89.68 ± 0.19[acdef] |
| Tiger | ≥96[d] | 0.18 ± 0.01[bc] | 9.26 ± 0.06[c] | 1.18 ± 0.05[g] | 148.19 ± 0.94[bcde] | 1.32 ± 0.12[c] | 1.99 ± 0.05[i] | 87.50 ± 0.60[h] |

| | YAN (mg/L) | | pH | | pH Decrease Rate (pH/h) | Specific Gravity | |
|---|---|---|---|---|---|---|---|
| Fermentation Time (h): | 0 | 96 | 0 | 96 | 24–48 | 0 | 96 |
| MR-Apollo | 131.51 ± 4.79[ac] | <DL | 4.25 ± 0.04[ad] | 3.36 ± 0.03[a] | 0.0144 ± 0.0037[acde] | 1.0881 ± 0.0005[j] | 0.9942 ± 0.0002[a] |
| Rippa | 123.24 ± 1.53[cd] | <DL | 4.13 ± 0.02[f] | 3.36 ± 0.03[a] | 0.0132 ± 0.0034[cde] | 1.0944 ± 0.0035[bd] | 0.9918 ± 0.0005[bc] |
| MR-Bazley | 116.68 ± 5.54[bdf] | <DL | 4.23 ± 0.06[d] | 3.30 ± 0.08[bdf] | 0.0156 ± 0.0006[acd] | 1.0920 ± 0.0003[bc] | 0.9925 ± 0.0005[b] |
| MR-Buster | 117.97 ± 8.60[d] | <DL | 4.25 ± 0.02[de] | 3.27 ± 0.02[f] | 0.0169 ± 0.0005[a] | 1.0943 ± 0.0010[bef] | 0.9925 ± 0.0003[b] |
| 84G22 | 116.28 ± 7.38[dg] | <DL | 4.28 ± 0.02[db] | 3.30 ± 0.02[ef] | 0.0151 ± 0.0005[acf] | 1.0961 ± 0.0020[deg] | 0.9917 ± 0.0001[bd] |
| 85G33 | 107.30 ± 7.44[efgh] | <DL | 4.29 ± 0.03[abe] | 3.27 ± 0.02[f] | 0.0167 ± 0.0008[ac] | 1.0950 ± 0.0010[bg] | 0.9917 ± 0.0003[be] |
| Liberty | 106.79 ± 2.94[fgi] | <DL | 4.18 ± 0.03[cg] | 3.34 ± 0.02[acde] | 0.0124 ± 0.0034[dfg] | 1.0943 ± 0.0025[bgh] | 0.9913 ± 0.0009[cdef] |
| MR-Taurus | 102.78 ± 2.58[hi] | <DL | 4.28 ± 0.02[abe] | 3.38 ± 0.04[a] | 0.0144 ± 0.0010[acgh] | 1.0913 ± 0.0014[acfhi] | 0.9918 ± 0.0002[bfh] |
| Cracka | 133.80 ± 9.64[a] | <DL | 4.18 ± 0.03[bg] | 3.29 ± 0.04[cf] | 0.0119 ± 0.0009[befh] | 1.0931 ± 0.0022[bgi] | 0.9919 ± 0.0002[bfg] |
| 85G44 | 119.10 ± 5.36[d] | <DL | 4.13 ± 0.04[f] | 3.27 ± 0.02[f] | 0.0088 ± 0.0026[bi] | 1.0943 ± 0.0007[abg] | 0.9913 ± 0.0005[cdegh] |
| Tiger | 100.24 ± 5.96[hi] | <DL | 4.26 ± 0.02[de] | 3.35 ± 0.02[ade] | 0.0083 ± 0.0004[i] | 1.0916 ± 0.0008[ab] | 0.9937 ± 0.0007[a] |

<DL = below detection limit; different letters within each column indicate significant difference in value ($p < 0.05$).

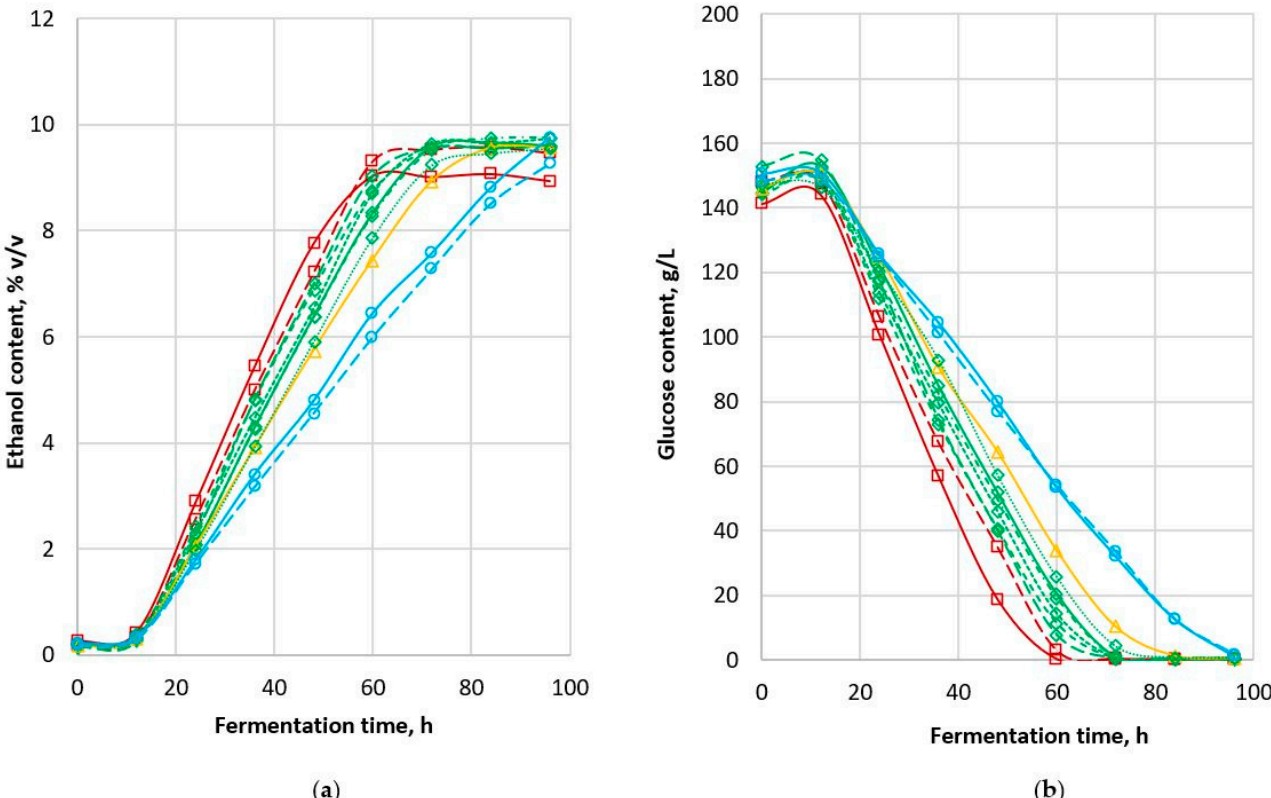

**Figure 1.** Changes in (**a**) ethanol content and (**b**) glucose content during 96 h of dry grind fermentation of 11 commercial sorghum cultivars (⬜ MR-Apollo, ⬜ Rippa, ◇ MR-Bazley, ◇ MR-Buster, ◇ 84G22, ◇ 85G33, ◇ Liberty, ◇ MR-Taurus, △ Cracka, ⊖ 85G44, ⊖ Tiger).

The final alcohol content of the wort was positively correlated with grain starch content (r = 0.736, *p* < 0.01). Since ethanol is a product produced by yeast from glucose, and glucose is the monomer released from starch breakdown, it is not surprising that grain samples with higher initial starch content had a higher final ethanol content after 96 h of fermentation. Higher ethanol yield from samples with higher starch content indicates better overall fermentation efficiency on a grain weight basis. The final ethanol content was negatively correlated with grain protein content (r = −0.885, *p* < 0.01) due to the inverse relationship between protein and starch content. These findings are consistent with those of Wu et al. (2007), who reported that starch content was positively correlated with ethanol yields (*p* < 0.001), while higher protein, crude fibre and ash contents had negative effects on ethanol yields (*p* < 0.001) [10]. The higher r value for the protein and ethanol content correlation suggests that protein content may be a more useful predictor of final ethanol content than starch content.

Ethanol production rate differed significantly between sorghum cultivars, ranging from 1.18 to 2.04 mL of ethanol per litre wort per hour of fermentation (Table 2). The ethanol production rates of MR-Apollo and Rippa were significantly higher than those of 85G44 and Tiger and the fermentations were completed more than 30 h earlier. This is unlikely to be due to differences in starch structure or starch digestibility affecting the release of glucose since the samples were pretreated with external enzymes to ensure glucose availability before fermentation. While MR-Apollo was one of the cultivars with the highest fermentation rate, its final ethanol content was the lowest of all the sorghums tested. In contrast, both 85G44 and Tiger had the lowest fermentation rate among all the cultivars tested, with ethanol contents of 9.75 and 9.26% *v/v*, respectively. Pearson

correlations confirmed that there was no relationship between ethanol production rate and ethanol content.

Ethanol production rate is known to play an important role in ethanol production capacity, thereby affecting the economy of the fermentation process. However, this may not be an issue when extrapolated to traditional Baijiu production, for which the fermentation takes about one month to complete. Moreover, traditional Baijiu fermentation is solid-state, which has a different mechanism than the dry grind fermentation used in this study. Solid-state fermentation of Baijiu is now the subject of intense application of science and engineering to improve its performance [21].

### 3.2.2. Consumption of Glucose

The initial wort glucose content for all the sorghum cultivars in the current study was consistently in the range 141–153 g/L (Figure 1b) (Table 2). The growth and proliferation of yeast cells require glucose as a substrate. However, there was no significant change in the glucose content observed in the first 12 h of fermentation. After the initial saccharification process, the further breakdown of starch into glucose in the first 12 h of fermentation and the concomitant consumption of glucose by yeast resulted in no change in overall glucose concentration. This was confirmed by a separate experiment using a similar setup without the presence of yeast. The starch-to-glucose conversion rate after saccharification was less than 90%. The conversion rate increased for all the samples consistently after 12 h incubation. This observation is consistent with the results of du Preez et al. (1985) who reported that saccharification was only 85–89% after 2 to 5 h saccharification using amyloglucosidase [22].

The glucose level of all sorghum cultivars decreased at different rates from 12 h onwards until the content was exhausted upon the completion of fermentation (Figure 1b). A brief deceleration phase was observed close to the endpoint. Fermentation is considered complete when the glucose content is exhausted. This study shows the final glucose content at a minimum level of 0.33–1.32 g/L, while the ethanol content was in the range 8.94–9.76% *v/v*, suggesting that glucose content might be the limiting factor in this fermentation. Thus, it is likely that an increment of fermentation substrate would have increased the final ethanol content, particularly as the yeast used in this study was a high-alcohol-tolerance type (up to 18%). In contrast, Nuanpeng et al. (2018) reported a final glucose content of 75 g/L and a final ethanol content of close to 100 g/L (equivalent to 12.7% *v/v*) at the fermentation endpoint, the high level of glucose residue, suggesting a high ethanol content that would inhibit further fermentation by the yeast [23].

The initial wort glucose content was not correlated with grain starch content or wort final ethanol content. This is mainly due to the incomplete breakdown of starch to glucose during the saccharification process (as discussed above). Protein bodies in the whole sorghum grain contain a high proportion of kafirins (70–80%), the main storage proteins of sorghum grain [24]. The interaction between kafirins and starch bodies may compromise the digestibility of starch [25]; however, in the current study, the initial wort glucose content was not correlated with grain protein content.

The rates of glucose consumption during the fermentation differed significantly between cultivars (Table 2) in the range 1.98–3.41 g per litre wort per hour of fermentation. The glucose consumption rate was strongly correlated with the ethanol production rate (r = 0.991, $p < 0.001$). Sorghum cultivars with high glucose consumption rates had high ethanol production rates, consistent with the results of Ratnavathi et al. (2010) on the fermentation of sorghum stalks [26].

### 3.2.3. Fermentability

Sample fermentability is a measure of the glucose-to-ethanol conversion rate. The fermentability of all 11 cultivars studied was in the range 87.3–89.7% (Table 2), consistent with a study by Wu et al. (2007) who reported an 86.3% conversion rate for low-ethanol-yield samples and a 90.2% conversion rate for high-ethanol-yield samples [10]. Mamma et al. (1995)

reported a relatively low maximum conversion rate of 78.6% [17], which could be due to the low ethanol tolerance threshold of the yeast used (~9%).

All sorghum samples used in this study were of low-tannin type (<0.25% catechin equivalent). Corredor et al. (2006) reported that the presence of tannin had negative effects on fermentation efficiencies due to the interference of starch breakdown into glucose for fermentation by the yeast. The ethanol fermentation of decorticated samples showed an improvement in the fermentation efficiencies of high-tannin sorghum samples to 87% [27]. Other factors such as the physical form of the starch granule and the extent of starch-protein interactions may also affect the digestibility of sorghum starch, thus affecting fermentation efficiency. Saccharification of starch was not an issue in this study as all the sorghum samples were treated with enzymes before fermentation to produce glucose.

Fermentability was positively correlated with sorghum starch content ($r = 0.676$, $p < 0.05$) and negatively correlated with protein content ($r = -0.775$, $p < 0.01$). A higher protein content of a sorghum cultivar suggests a possible higher content of kafirin and starch complexes that have lower digestibility and hence lower fermentability. However, this is inconsistent with the results of a study by Wu et al. (2007) in which neither starch content nor protein content was correlated with glucose-to-ethanol conversion rate [10]. This may be due to the consistency of samples used in that study, whereas in the current study, all 11 cultivars were planted in a single location and under the same management, which enabled valid comparisons between cultivars.

### 3.2.4. Consumption of Yeast Assimilable Nitrogen (YAN)

The initial YAN content of all cultivars was 100–134 mg/L, which decreased consistently after 12 h (Figure 2; Table 2). YAN is required (in addition to glucose) by the yeast to proliferate. Within 24 h of fermentation, the YAN content of 85G33 and Liberty was below the detection limit (20 mg/L) while the YAN content of the other nine cultivars was depleted to near the detection limit. From 36 to 48 h of fermentation, YAN was detected only in Cracka and 85G44. The YAN content of all cultivars was below the detection limit from 60 h onwards.

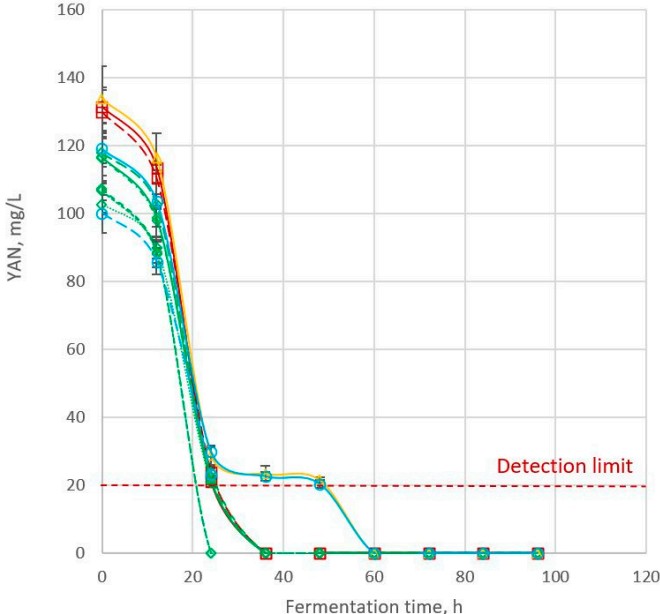

**Figure 2.** Changes in yeast assimilable content during 96 h of dry grind fermentation of 11 Australian sorghum cultivars ( MR-Apollo, Rippa, MR-Bazley, MR-Buster, 84G22, 85G33, Liberty, MR-Taurus, Cracka, 85G44, Tiger).

YAN content was calculated from ammonia content and free amino nitrogen (FAN) content. Bvochora et al. (2000) analysed only FAN content to indicate the level of fermentation nutrients but reported the same trend as observed here: an initial decrease in FAN up to 24 h, then depletion of FAN, which remained at a minimum level for the rest of the fermentation process [19]. This is different from the very high-gravity fermentation studied by the same group, where the FAN similarly decreased up to 36 h.

YAN content was not correlated with grain protein content, indicating that not all nitrogen available in the sorghum grain was accessible by yeast. Interestingly, the initial YAN content (at 0 h) was also not correlated with ethanol content or ethanol production rate. There was no clear trend of initial YAN content on ethanol content or ethanol production rate; i.e., the cultivars with high YAN did not give high final ethanol content or high ethanol production rate. The correlation reported in this study was determined using experimental values for 11 different sorghum cultivars. However, if the YAN content for one particular cultivar were increased, there would likely be a higher fermentation rate. Previous studies have shown that increased concentrations of YAN facilitated yeast fermentation of sugar to ethanol [10,28].

3.2.5. Other Changes during Fermentation

The conversion of glucose to ethanol during fermentation resulted in changes in pH. The pH value of all samples remained around 4.1–4.3 in the first 12 h of fermentation (Figure 3; Table 2). Sample pH then decreased sharply in the 12–24 h period of the fermentation to pH 3.6–3.8. The decrease continued until the fermentation endpoint, after which the pH of all samples increased slightly. All cultivars except Tiger and 85G44 had minimum pH values of 3.1–3.3 close to the fermentation endpoint (60–72 h), but then the pH increased gradually to pH 3.3–3.4. This trend is consistent with the results reported by Nuanpeng et al. (2018) even though the actual pH values were different due to the different fermentation method and microorganism used [23]. The initial pH reported by these workers was 5.4 while the final pH after 84 h of fermentation was 4.7 [23]. In the current study, Tiger and 85G44 displayed trends in which the pH decreased slowly and continuously from 0 to 96 h of fermentation. Whether the pH would increase or decrease further after 96 h was not known since the fermentation was stopped at that time point and no data were collected beyond this point.

The decrease in pH described above may be due to the initial absorption of basic amino acids present in the wort [29], followed by the secretion of organic acids, the dissolution of $CO_2$ and the absorption of primary phosphate as the fermentation progressed. The uptake and consumption of glucose by different sorghum cultivars in this study occurred at different rates (Table 2); hence the difference in the fermentation rate, and the production rate of acidic metabolites that decreased the pH value. The rate at which the fermentation system pH decreased consistently followed the trend of glucose consumption rate and ethanol production rate. The samples with a high rate of pH decrease had a high glucose consumption rate and high ethanol production rate (r = 0.838, $p < 0.001$; r = −0.863, $p < 0.001$, respectively).

The balance of glucose content and ethanol content during fermentation was reflected in the specific gravity profile of the samples. Sorghum samples with decreasing glucose content and increasing ethanol content during fermentation had a changing SG profile (Figure 4). Initial specific gravity (at 0 h) for all cultivars was in the range 1.088–1.096 (Table 2). The specific gravity decreased to a minimum level of 0.991–0.994 with Apollo reaching the minimum specific gravity first with Tiger last. The specific gravity then remained constant after the fermentation endpoint. The decrease in specific gravity over time is mainly due to the consumption of glucose and the concomitant production of ethanol, both of which lowered the solution density. This result is consistent with the findings of Bvochora et al. (2000) and Nuanpeng et al. (2018), which showed changes in the concentration of dissolved solids with fermentation time [19,23]. The same trend was

observed in both studies as the dissolved solid content decreased and the alcohol content increased during fermentation.

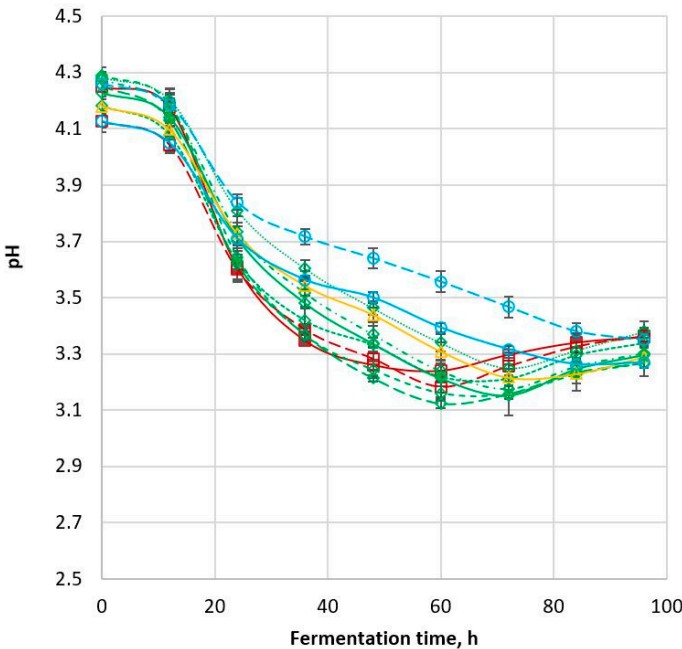

**Figure 3.** Changes in pH values during 96 h of dry grind fermentation of 11 commercial sorghum cultivars (⸺☐⸺ MR-Apollo, ⸺☐⸺ Rippa, ⸺◇⸺ MR-Bazley, ⸺◇⸺ MR-Buster, ⸺◇⸺ 84G22, ⸺◇⸺ 85G33, ⸺◇⸺ Liberty, ⸺◇⸺ MR-Taurus, ⸺△⸺ Cracka, ⸺○⸺ 85G44, ⸺○⸺ Tiger).

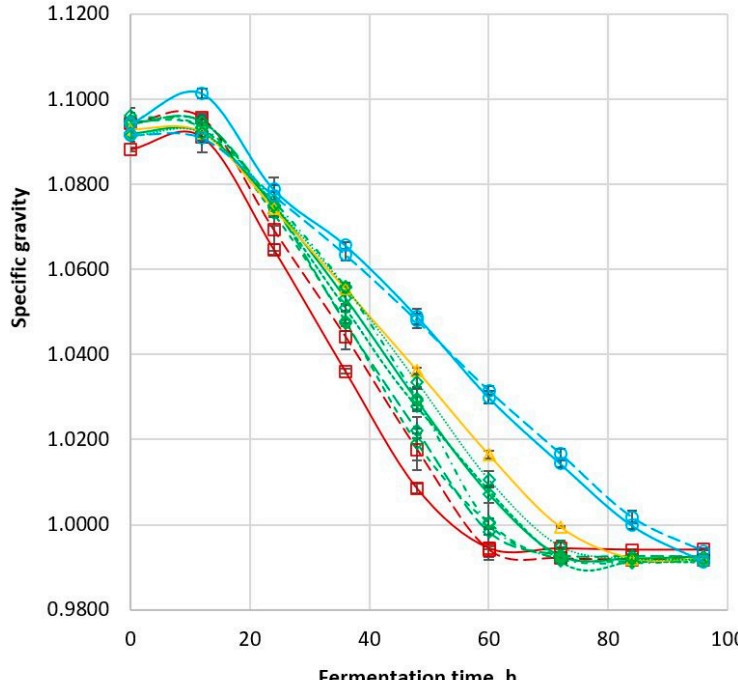

**Figure 4.** Changes in specific gravity during 96 h of dry grind fermentation of 11 commercial sorghum cultivars (⸺☐⸺ MR-Apollo, ⸺☐⸺ Rippa, ⸺◇⸺ MR-Bazley, ⸺◇⸺ MR-Buster, ⸺◇⸺ 84G22, ⸺◇⸺ 85G33, ⸺◇⸺ Liberty, ⸺◇⸺ MR-Taurus, ⸺△⸺ Cracka, ⸺○⸺ 85G44, ⸺○⸺ Tiger).

## 4. Conclusions

The current study determined the fermentation performance of grains from 11 Australian sorghum cultivars sourced from a field trial. We were unable to replicate the solid-state fermentation used in Baijiu production and instead performed a dry-grind fermentation, which is a substantial limitation of this study. However, we judged that our dry-grind fermentation experiments would allow Australian sorghum genotypes to be compared as a first step towards their later evaluation in a more realistic Baijiu production setting.

Results from the current study showed that these sorghums can be categorised into four groups according to the fermentation completion time of 60–69 h (MR-Apollo and Rippa), 70–79 h (MR-Bazley, MR-Buster, 84G22, 85G33, Liberty and MR Taurus), 80–89 h (Cracka) and >90 h (85G44 and Tiger). Apollo was the only cultivar that produced less than 9.0% ethanol content; all other cultivars produced 9.47–9.76% $v/v$ of ethanol in the final product. The final alcohol content of the wort was positively correlated with grain starch content (r = 0.736, $p < 0.01$) and negatively correlated with grain protein content (r = $-0.885$, $p < 0.01$).

Glucose consumption rate differed between cultivars and was strongly correlated with the ethanol production rate (r = 0.991, $p < 0.001$). Fermentability was positively correlated with sorghum starch content (r = 0.676, $p < 0.05$) and negatively correlated with protein content (r = $-0.775$, $p < 0.01$). There were no correlations between initial wort glucose content or YAN content with grain starch content, protein content, ethanol content, fermentability, ethanol production rate or glucose consumption rate. All grain samples used in this study were from non-waxy and low-tannin cultivars as these are the only sorghum types grown commercially in Australia. The relative performance of these genotypes in the dry grind fermentation conducted here will enable the selection of sorghums for performance testing in solid-state fermentations using authentic Baijiu cultures. Waxy and high-tannin genotypes of sorghum should be studied in the future as these are also used in China for commercial Baijiu production.

**Author Contributions:** S.H.T.: Methodology; Formal analysis; Investigation; Writing—original draft. C.L.B.: Project administration; Supervision; Writing—review and editing. T.H.R.: Writing—review, editing and submission. D.L.E.W.: Writing—review and editing; Supervision. A.J.M.: Conceptualization; Writing—review and editing. All authors have read and agreed to the published version of the manuscript.

**Funding:** This work was supported by the Grains Research & Development Corporation (GRDC), grant number UCS00025.

**Data Availability Statement:** The data presented in this study are available in the article.

**Acknowledgments:** The authors thank Helen Pan and Campbell Meeks of the National Wine and Grape Industry Centre (NWGIC) for their assistance in this study. We thank Pioneer, Nuseed and Pacific Seeds for the sorghum planting seed used for the field trial, as well as Annette Tredrea and coworkers (University of Sydney) for running the field trial.

**Conflicts of Interest:** The authors declare no conflict of interest. The funders had no role in the design of the study; in the collection, analyses, or interpretation of data; in the writing of the manuscript; or in the decision to publish the results.

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
