# Peer review of "Effects of Sorghum Grain and Wort Composition on Dry Grind Fermentation Performance: A Model for Baijiu Production"

_beverages, doi:10.3390/beverages9020029_

Round 1
Reviewer 1 Report
In this manuscript, it was similar to many results based on the influence of starch content in different varieties grain on ethanol yield, it has an important reference role for breeding high ethanol yield or conversion rate. However, the research scheme has the following defects in the design aiming technological characteristics of Chinese Baijiu. 1) There is a big difference between the experimental scheme and Baijiu technology. The latter is based on the synergistic effect of microbial flora and complex enzymes on sorghum hydrolysis.
2) The ethanol fermentation of Chinese Baijiu also depends on a variety of bacteria.
Therefore, it is worth discussing whether the research results only focusing on the ethanol conversion rate of starch and protein can be used for reference in Baijiu brewing
Author Response
We are grateful for the reviewer’s questions.
1). We agree that there is a big difference between our experimental scheme and the processes involved in Baijiu technology. Our scheme serves as an initial but substantial step towards understanding the potential of Australian sorghum grain (comprising 11 cultivars) as a fermentation substrate for Baijiu production. We are well aware of the limitations of dry grind fermentation as a model for the biologically complex process required to make Baijiu.
2). We have now added some text to the Conclusions discussing the relevance of our findings for Baijiu production. Thus, in lines 390-392 of the revised manuscript, we write, ‘The relative performance of these genotypes in the dry grind fermentation conducted here will enable selection of sorghums for performance testing in solid-state fermentations using authentic Baijiu cultures.’
Reviewer 2 Report
Dear authors, after reviewing the research article "Effects of Sorghum Grain and Wort Composition on Dry Grind Fermentation Performance" I would like to comment that I appreciate the care for the writing and presentation of the article, it is precise in the explanation and clear in the graphics. I have only one doubt, sorghum grains were decorticated before milling? according to the reading it was not so, if confirmed, what effect could have the dietary fiber and germ on the experiment? it would be worth to analyze this aspect in the future.
Best regards
Author Response
We greatly appreciate the reviewer’s positive remarks on our writing and presentation and the insightful questions. The sorghum grains were not decorticated. We have now made that clear in the second sentence of Section 2.3 Dry Grind Fermentation (line 81 of the revised manuscript) by inserting the word ‘Whole’ before ‘sorghum grain’. Our understanding is that Baijiu production is normally performed using whole-grain sorghum, such that the bran layers (high in fibre) and the germ (containing substantial lipid) are present in the fermented grain. We agree with the reviewer that the effects of the bran and germ on dry grind fermentation performance of sorghum grain are worthy of investigation in the future.
Reviewer 3 Report
The authors conducted interesting research about the effects of sorghum grain and wort composition on dry grind fermentation performance.
I suggest inserting the name of the beverage into the title. The current title is a bit misleading in the sense that it is not clear which beverage it refers to.
Otherwise, I have no remarks.
Author Response
We thank the reviewer for suggesting a change to the manuscript title. We have now extended the title to the following: Effects of Sorghum Grain and Wort Composition on Dry Grind Fermentation Performance: A Model for Baijiu Production.
Reviewer 4 Report
In this manuscript the authors investigated the effects of sorghum grain quality on fermentation performance for Baijiu production, using grains from a selection of 11 sorghum cultivars (grown in a field trial in Australia).
The production technology of Baijiu outside China has important engineering value as the internationalization process of Baijiu is very limited. Thus, this is an interesting topic and the purpose of the manuscript is valuable.
The paper can be considered to be published in Beverages after a minor revision.
The minor issues should be addressed:
1. The scientific and engineering significance of the paper remains unclear in the present form of the manuscript, the author needs to strengthen the Introduction section.
2. As stated by the author, commercial Baijiu is produced with a solid-state fermentation process. However, in this study, fermentation of sorghum under dry grind conditions was carried out with a liquid fermentation process. The reasons for this need to be stated.
Author Response
We thank the reviewer for asking two valuable questions.
1). We have now added some text to the Introduction to strengthen our explanation of the significance of the paper. Thus, in lines 59-62 of the revised manuscript, we write, ‘The significance of this study is that it establishes relationships between quality parameters of non-waxy, low-tannin sorghum grain and dry grind fermentation performance. The results can be used to inform research to optimise grain quality for solid-state fermentation in Baijiu production.’
2). The liquid fermentation process used in our study is a simple model of the much more complex process used in commercial Baijiu production. Our aim here was to compare the fermentation potential of the major Australian sorghum genotypes, which would inform the selection of cultivars suitable for further testing with Baijiu production processes. That is now clear in the text we have now added in response to the reviewer’s first question.
Round 2
Reviewer 1 Report
More reasonable modifications have been made, especially the modification of the title, which gives the research results great significance
Author Response
We wish to thank the reviewer for confirming the value of our altered manuscript title.